# Individual and community-level determinants of underweight among lactating mothers in Ethiopia: A multilevel analysis

Zinash Teferu[1]*, Yohannes Tekalegn[1], Biniyam Sahiledengle[1], Demisu Zenbaba[1], Fikreab Desta[1], Kenbon Seyoum[2], Habtamu Gezahegn[3], Damtew Solomon Shiferaw[4], Ayele Mamo[5], Vijay Kumar Chattu[6,7]

1 Department of Public Health, Madda Walabu University Goba Referral Hospital, Bale-Goba, Ethiopia,
2 Department of Midwifery, Madda Walabu University Goba Referral Hospital, Bale-Goba, Ethiopia,
3 Physiology Department, Madda Walabu University Goba Referral Hospital, Bale-Goba, Ethiopia,
4 Anatomy Department, Madda Walabu University Goba Referral Hospital, Bale-Goba, Ethiopia,
5 Pharmacy Department, Madda Walabu University Goba Referral Hospital, Bale-Goba, Ethiopia, 6 Division of Occupational Medicine, Department of Medicine, Temerty Faculty of Medicine, University of Toronto, Toronto, ON, Canada, 7 Department of Public Health, Saveetha Medical College and Hospitals, SIMATS, Saveetha University, Chennai, India

* zinut2016@gmail.com

**Data Availability Statement:** Data are available with permission from the Measure DHS website (http://www.dhsprogram.com).

## Abstract

### Background

Determining the nutritional status of lactating women is important because underweight lactating mothers will have low energy levels and reduced cognitive abilities, which will affect the inadequate care of their young children. Thus, malnutrition is passed down from generation to generation, perpetuating the vicious cycle. There is scarce national data on determinants of underweight among lactating mothers in Ethiopia. Hence, this study aimed to identify individual and community-level determinants of underweight among lactating mothers in Ethiopia.

### Methods

Data from the Ethiopian Demographic and Health Survey (EDHS) from 2016 were used. A total of 3848 lactating mothers were included in this study, and a multilevel, multivariable logistic regression model was fitted to identify determinants of underweight among lactating mothers.

### Results

The odds of being underweight among rural lactating mothers were 65% higher (AOR = 1.65, 95% CI = 1.13, 2.41) than lactating mothers in the urban area. The odds of being underweight among lactating mothers who have toilet facilities were 33% lower (AOR = 0.67, 95%CI = 0.54, 0.83) compared with those do not have toilet facilities. Those mothers in the age group of 25–34 years and greater than 35 years had (AOR = 0.61,95%CI = 0.48, 0.79), and (AOR = 0.66, 95% CI = 0.47, 0.95) times lower chance of being underweight

**Funding:** The author(s) received no specific funding for this work.

**Competing interests:** The authors have declared that no competing interests exist.

compared with those who had 15–24 years of age, respectively. The likelihood of being underweight among lactating mothers in high community poverty (AOR = 1.40, 95%CI = 1.08, 1.82) was higher than the lower community poverty level.

## Conclusion

Underweight among lactating mothers was significantly associated with individual-level (age and toilet facilities) variables and community-level (residence and community poverty). Therefore, focusing on these identified factors could improve underweight among lactating mothers in Ethiopia.

## Background

Maternal underweight is defined as having a body mass index of less than 18.5 kg/m$^2$ [1]. Maternal underweight is still common and associated with millions of deaths each year [2]. Globally, maternal and child under-nutrition causes 3.5 million deaths every year and 11% of total global disability-adjusted life years [3]. Undernutrition is one of the common health problems affecting millions of people in developing countries and contributes to poor health and nutritional status leading to chronic energy deficiency [4]. The nutritional status of pregnant and lactating women is very important since it also affects their children's health [5–9].

The physiological pressure of added nutrient demands makes high-risk lactation periods in a mother's life [10]. The extra energy needed during lactation represents 8% of total household food energy needs [11]. Lactating moms have a larger nutritional demand, and if it is not adequately met, breast milk composition and production are negatively affected leading to increased risk for child morbidity and mortality [4, 9].

Underweight lactating mothers will have low energy levels and reduced cognitive abilities, which will affect the inadequate care of their young children [12]. In this way, underweight is passed down from one generation to another, and the vicious cycle continues. These children do not experience much catch-up growth in subsequent years, remain susceptible to diseases, enter school late, do not learn well, and are less productive as adults [13].

Many studies have identified age of the mother, educational status of mother, place of residence, region [14–18], family size, community nutrition program [19, 20], household toilet facility [15, 21, 22], household wealth index [17], and birth to pregnancy interval and postnatal care [18, 22] as significant factors that determine of being underweight among lactating mothers.

The poor nutritional status of lactating women is considered one of the greatest threats to the world's public health and is a serious developmental threat to many developing countries [23, 24]. Studies show a significantly high prevalence of underweight lactating women in Ethiopia, the highest proportion of underweight lactating mothers in Sub-Saharan Africa [24].

The Ethiopian government has developed many programs, policies, and initiatives to improve maternal underweight in Ethiopia. The National Nutrition Program, the Health Sector Development Plan IV, the 2015 Seqota declaration, the 2015–2020 Productive safety net program, and the Agricultural growth program are among the programs [25]. Regardless of this effort in Ethiopia, studies have shown a higher prevalence of underweight among lactating mothers [26]. Another Ethiopian review study has identified the highest prevalence at 50.6% in the Northern parts and the lowest at 17.4% in the Southern parts of Ethiopia [19].

Even though being underweight among lactating mothers is high, there is scarce national data on determinants of underweight among lactating mothers. In Ethiopia, A few studies were conducted to identify associated factors of underweight among lactating mothers. More-over, a previous study was specifically localized to areas of our country with small size, and therefore their findings did not represent the entire nation [15, 16, 18, 20]. In addition, the influence of communities (community poverty level, place of residence, and regions) on being underweight among lactating mothers was overlooked.

Therefore, this study is intended to identify individual and community-level effects on underweight among lactating mothers in Ethiopia using a multilevel approach. The study's findings will assist health policy experts and decision-makers develop more effective and com-prehensive intervention programs, in which treatments are targeted at the individual and com-munity levels, allowing for possible long-term structural changes and effective policymaking.

## Methods

### Data source and sampling procedures

Data from the 2016 Ethiopian Demographic and Health Survey were used. The EDHS was cross-sectional by design, and data had been collected by the Ethiopian Central Statistical Agency (ECSA) from January 18, 2016, to June 27, 2016. The research was carried out in accordance with the STROBE cross-sectional reporting guidelines (**S1 Table**).

Administratively, Ethiopia is divided into nine geographical regions and two administrative cities. Each district is further subdivided into the lowest administrative unit, called kebeles. Kebeles are also subdivided into census enumeration areas (EAs), convenient for implement-ing the survey. A stratified, two-stage cluster sampling design was employed. The sampling frame used for the EDHS-2016 is the Ethiopia Population and Housing Census (PHC), con-ducted in 2007 by the ECSA. The census frame is a complete list of 84,915 enumeration areas (EAs) created for the 2007 PHC. We have used EAs as primary sampling units (PSU) and households as the secondary sampling units. The EDHS- 2016 include 645 EAs, 202 in urban and 443 in rural areas. In the second stage of selection, a fixed number of 28 households per cluster were selected with an equal probability of systematic selection from the newly created household listing. In the current study, we included 3848 lactating mothers as well as commu-nity characteristics of 612 clusters. The lactating mothers who were pregnant or who gave birth in the two (2) months preceding the interview date were excluded from the study.

The EDHS-2016 data sets were downloaded in STATA format with permission from the Measure DHS website (http://www.dhsprogram.com).

### Study variables

**Outcome variables.**    The outcome variable for this study was the nutritional status of lac-tating mothers and was derived from body mass index (BMI), which is dichotomized as nor-mal and underweight. Lactating mothers are found to have normal BMI when their BMI is from 18.5 to 24.9 kg/m2. Those lactating mothers with a BMI of $<18.5$ kg/m$^2$ were under-weight. According to EDHS data, height and weight measurements were carried out on women aged 15–49 years in all selected households. Weight measurements were obtained using lightweight SECA mother-infant scales with a digital screen designed and manufactured under the guidance of UNICEF. Height measurements were carried out using a Shorr measur-ing board.

**Independent variables.**    The independent variables for this study were classified as indi-vidual/ household level and community level factors. The selection of independent variables was based on reviewed literature [15, 17–20]. The individual/household level factors were age,

marital status (labeled as married or living with partner and other), maternal occupation (working or not working), parity (1–2,3–6, ≥7), maternal education (no education, primary education, secondary education, and higher education), wealth index (poor, middle and rich), alcohol drink (yes, no), religion (orthodox, protestant, muslim and other), birth interval (≤23 months, ≥24 months), postnatal care (yes, no), community conversion (yes, no), media exposure (yes, no), toilet facility (yes, no), hand washing facility (yes, no), source of drinking water (improved, unimproved), chat chewing (yes, no), and contraceptive use (yes, no). The community-level factors included: the place of residence (urban or rural), contextual region (agrarian, pastoralist, and cities), and Level of "poor" households in the community was defined as the proportion of mother in poorest and poorer households within each cluster. The household wealth index variable was used, which indicates a household's cumulative living standard. In the EDHS data, the majority of mothers are poor. This variable was categorized as high poverty or low poverty using a median split. Then the proportion of high poverty households for each cluster was calculated. The last one, community-level factors are created by aggregating the selected household-level factors at the cluster level (not directly found in the Demographic Health Survey data). The aggregates were computed using the average values of the proportions of women in each category of a given variable. We used median values to categories the aggregated variables into groups.

## Data management and analysis

The data were checked for completeness and weighted before doing any statistical analysis. The analysis was done using STATA Version 14. Due to the non-proportional allocation of the sample to different regions, urban and rural areas, and the possible differences in response rates, proportions and frequencies were estimated after applying sample weights to the data to account for disproportionate sampling and nonresponses, recommended by the EDHS. A detailed explanation of the weighting procedure can be found in the EDHS report [14].

Since EDHS data are hierarchical, i.e., mothers are nested within households, and households are nested within clusters, the use of standard models could underestimate standard errors of the effect sizes, consequently affecting the decision on the null hypothesis. With such data, mothers within a cluster may be more similar to each other than mothers in the rest of the cluster. This violates the assumption of the standard model; independence of observation and equal variance across the cluster which implies a need to consider the variability between clusters. All these issues are motivated to use the multilevel modeling, which was able to compute mixed effect that fixed effect for both individual/household and community factors of underweight and a random effect for between cluster variations simultaneously. As the response variable, underweight was dichotomous multilevel logistic regression was used.

Both bi-variable and multi-variable multilevel logistic regression was performed to identify the determinants of underweight. All variables with a p<0.25 at bi-variable multilevel logistic model analysis were entered into the multivariable multilevel logistic regression model. Variance inflator factor (VIF) was employed for checking multicollinearity among the independent variables. The fixed effect sizes of individual/household and community level factors on underweight were expressed using the adjusted odds ratios (AOR) with a 95% confidence interval (CI). A p-value of <0.05 was used to declare statistical significance. The random effects (variation of effects) were measured by ICC, PCV, and median odds ratio (MOR), which measure the variability between clusters in the multilevel model [27–29]. The ICC explains the cluster Variability, while MOR can quantify unexplained cluster variability (heterogeneity). MOR translates cluster variance into OR scale. In the multilevel model, PCV can measure the total variation due to factors at the community and individual levels [28]. The ICC, PCV, and MOR

were determined using the estimated variance of clusters using the following formula ICC = $\frac{V}{V+\frac{\pi^2}{3}}$ Where V denotes community variance, and $\frac{\pi^2}{3}$ denotes individual-level variance that is fixed for log distribution (equal to 3.29).

MOR = $\exp(\sqrt{2 \times V} \times 0.6745) \approx \exp(0.95\sqrt{V})$ Where V is the estimated variance of clusters and PCV = $\left(\frac{Va-Vb}{Va}\right) \times 100$ Where Va = variance of the initial (null) model; Vb = variance of the model with more terms [27, 28].

Model comparison was done using Akaike information criteria (AIC) and Bayesian information criteria (BIC). The comparison was done among the null model (a model with no independent variables), model I (a model with only individual/household-level factors), model II (a model with only community-level factors), and model III (a model with both individual and community-level factors). A model with the lowest AIC and BIC (model III) was selected.

**Ethical consideration.** The Ethiopian Demographic and Health Survey was conducted after the approval of the Ethiopian National Research Ethics Review Committee. Permission to use the 2016 EDHS database for further analysis was sought from http://www.dhsprogram. com and no ethics committee approval was necessary. The data was analyzed and reported in aggregate; household and individual identifiers were not reported in the dataset.

## Results

### Socio-demographic characteristics participants

A total of 3848 lactating mothers were included in this study. Of these, 22.7% (95 CI: 21.4%-24.1%) were undernutrition. About half (50.2%) of the mothers were aged 25–34 years. Of all mothers, 3,647 (94.8) were married or living with a partner, and 2,640 (68.6%) of the mothers did not have media exposure. A large proportion of the mothers (64.3%) had no formal education, while only a minimal 3.1% had attended higher school. Nearly half (53.5%) of the mothers were not currently working. About 45.1% of the participants were in poor household wealth quintiles (**Table 1**).

**Community-level variance and model comparison.** In the null model, community-level variance analysis was performed. Community-level variance indicates the total variance of under-weight among lactating mothers that can be attributed to the community (cluster) context in which the mother resided.

The use of a multilevel logistic regression model in the analysis was justified by the significance of the community-level variance [community variance = 0.55; standard error (SE) = 0.074; P-value = 0.001], indicating the existence of significant differences between communities (clusters) regarding the underweight among lactating mothers. The intra-cluster correlation coefficients (ICC = 0.144) supported this, revealing that 14.4 percent of the total variance of lactating mothers in Ethiopia could be attributable to the context of the communities in which the mothers lived. After adjusting for individual-level and community-level factors, the variation underweights among lactating mother communities remained statistically significant. About 7.2% of the odds of underweight variation across communities were observed in the full model.

**Table 2** (Model 4) also revealed that the PCV for individual and community-level factors model adjustment was 53.7%, indicating that 53.7% of the variance in the odds under-weight between communities was explained by individual and community-level factors found in the model. Moreover, median OR (MOR) for underweight was 2.02 in the null model, indicating heterogeneity between clusters. If we randomly select a mother from two different clusters, individuals at the cluster with a higher risk of underweight had 2.02 times higher odds of being underweight than individuals at cluster with a lower risk of underweight.

**Table 1. Underweight among lactating mothers by background individual and community-level characteristics in Ethiopia, 2016 (n = 3,848).**

| Variables | Weighted frequency(n) | Weighted percent (%) |
|---|---|---|
| **Individual-level variables** | | |
| **Age of the mother** | | |
| 15–24 years | 1022 | 26.6 |
| 25–34 years | 1933 | 50.2 |
| ≥35 years | 893 | 23.2 |
| **Religion** | | |
| Orthodox | 1530 | 39.8 |
| Protestant | 882 | 22.8 |
| Muslim | 1324 | 34.6 |
| Other | 112 | 2. 8 |
| **Marital status** | | |
| Married or living with a partner | 3647 | 94.8 |
| Other* | 201 | 5.2 |
| **Education of women** | | |
| No education | 2473 | 64.3 |
| Primary | 1135 | 29.5 |
| Secondary | 169 | 4.4 |
| Higher | 71 | 1.8 |
| **Wealth index** | | |
| Poor | 1734 | 45.1 |
| Medium | 907 | 23.5 |
| Rich | 1207 | 31.4 |
| **Employment status** | | |
| Not working | 2060 | 53.5 |
| Working | 1788 | 46.5 |
| **Husband education(n = 3,647)** | | |
| No education | 1734 | 47.6 |
| Primary | 1536 | 42.1 |
| Secondary | 263 | 7.2 |
| Higher | 114 | 3.1 |
| **Birth interval (n = 3,087)** | | |
| ≤23 month | 500 | 16.2 |
| ≥24 month | 2587 | 83.8 |
| **Contraceptive use** | | |
| No | 2323 | 60.4 |
| Yes | 1525 | 39.6 |
| **Parity** | | |
| 1–2 | 1353 | 35.2 |
| 3–6 | 1770 | 46.0 |
| ≥7 | 725 | 18.8 |
| **PNC** | | |
| No | 3611 | 93.8 |
| Yes | 237 | 6.2 |
| **Community conversion** | | |
| No | 2218 | 57.6 |
| Yes | 1630 | 42.4 |

(*Continued*)

**Table 1.** (Continued)

| Variables | Weighted frequency(n) | Weighted percent (%) |
|---|---|---|
| **Media exposure** | | |
| No | 2640 | 68.6 |
| Yes | 1208 | 31.4 |
| **Toilet facility** | | |
| No | 1489 | 38.7 |
| Yes | 2359 | 61.3 |
| **Hand washing facility** | | |
| Yes | 2092 | 54.4 |
| No | 1756 | 45.6 |
| **Source of drinking water** | | |
| Improved source of drinking water | 2217 | 57.6 |
| Unimproved source of drinking water | 1631 | 42.4 |
| **Chat chewing** | | |
| No | 3313 | 86.1 |
| Yes | 535 | 13.9 |
| **Alcohol drink** | | |
| No | 2503 | 65.1 |
| Yes | 1345 | 34.9 |
| **Community-level variables** | | |
| **Region** | | |
| Agrarian | 3666 | 95.3 |
| Pastoralist | 116 | 3.0 |
| City | 66 | 1.7 |
| **Place of residence** | | |
| Urban | 349 | 9.1 |
| Rural | 3499 | 90.9 |
| **Community poverty level** | | |
| Low | 2344 | 60.9 |
| High | 1504 | 39.1 |

The models were compared with AIC, BIC, and log-likelihood. The AIC and BIC values of the Null model, Model-I, Model-II, and Model-III were 3788.497, 3800.672, 3548.417, 3687.176, 3739.148, 3781.759, and 3537.085, 3706.009, respectively (**Table 2**). Lower values indicate the goodness of fit of the multilevel model. The largest values of Log-likelihood, AIC, and BIC were observed in Model III, and this implies that Model-III for underweight mothers was a better explanatory model.

**Determinants of underweight among lactating mother.** Age, religion, mother's education, parity, contraception, wealth index, hand washing facility, community conversion, employ status, husband education, alcohol use, media exposure, residence, region, and community poverty level were all associated with underweight ($p < 0.25$) in a bivariable multilevel logistic regression analysis. However, in the final model: age, residence, toilet facility, and community poverty level were significantly associated with underweight among lactating mothers in Ethiopia ($p \leq 0.05$).

The odds of being underweight among rural lactating mothers were higher than lactating mothers in the urban area (AOR = 1.65, 95% CI = 1.13, 2.41). The odds of being underweight were lower among lactating mothers who have toilet facilities (AOR = 0.67, 95%CI = 0.54, 0.83) compared with no had toilet facility.

**Table 2. Community-level variance and model comparison of multilevel logistic regression model predicting under-weight among lactating mothers, Ethiopia 2016.**

| Measure variation | Null model | Model I | Model II | Model III |
|---|---|---|---|---|
| Community-level–variance (SE) | 0.55(0.074)*** | 0.27(0.08)*** | 0.40(0.07)*** | 0.25(0.08)*** |
| ICC% | 14.4 | 7.7 | 10.8 | 7.2 |
| PCV% | Reference | 50.1 | 27.41 | 53.7 |
| MOR | 2.02(1.83,2.21) | 1.64(1.37,1.87) | 1.82(1.61,2.02) | 1.61(1.32,1.85) |
| Model fit statistics | | | | |
| AIC | 3788.497 | 3548.417 | 3739.148 | 3537.085 |
| BIC | 3800.672 | 3687.176 | 3781.759 | 3706.009 |
| LL | -1892.2485 | -1751.2084 | -1862.5739 | -1740.5423 |

SE: Standard error; ICC: Intra-class Correction Coefficient; MOR: Median Odds Ratio; PCV: Proportional Change in Variance; AIC: Akaike's Information Criterion;

BIC: Bayesian Information Criteria; LL: Log-likelihood

Null model: a model with no independent variables

Model I: a model with only individual/household-level factors

Model II: a model with only community-level factors

Model III: a model with both individual and community-level factors

***p-value<0.001

Those mothers in the age group of 25–34 years and greater than 35 years had (AOR = 0.61,95%CI = 0.48, 0.79), and (AOR = 0.66, 95%CI = 0.47, 0.95) times lower chance of being underweight compared with those who had 15–24 years of age, respectively. The likelihood of being underweight among lactating mothers in high community poverty (AOR = 1.40, 95%CI = 1.08, 1.82) was higher than the lower community poverty level (**Table 3**).

## Discussion

A multilevel logistic regression model was employed to investigate the determinants of underweight among Ethiopian lactating mothers in this study. This paper aimed to understand the individual and community level determinants of underweight among lactating mothers in a developing country, Ethiopia. It was also found that being underweight among lactating mothers was significantly associated with different individual and community level factors. At the individual level, age and toilet facilities are associated with being underweight among lactating mothers. At the community level, residence and community poverty were the identified factors associated with being underweight among lactating mothers.

According to our results, 22.7% (21.4%, 24.1%) of lactating mothers in Ethiopia were underweight; the highest proportion was reported in rural areas. This is less than with previous related studies, which found underweight among lactating mothers in Ethiopia 50.6% [26], 26.1% [21], and 25.4% [20]. This disparity could be attributed to a difference in study participant timing, as previous studies were conducted during religious fasting season as religious fasting is one of the dietary or food taboos that can impact lactating mothers' dietary intake and nutritional status, resulting in undernourished breastfed children. The key difference between religious fasting and ordinary food taboos is that religious fasting is only temporary, requiring abstention from consuming animal source foods and/or certain foods [30]. In addition to this, evidence has found that Ramadan fasting affects women of reproductive age's nutritional status, dietary nutrient intake, birth outcome, breast milk content, and overall health [31–33]. Moreover, the other reason for the difference might be that our data is

**Table 3. Determinants of Underweight among lactating mother in Ethiopia, 2016; results for multilevel logistic models.**

| Variables | Under-weight | | Crude OR (95%CI) | Adjusted O(95%CI)[#] |
|---|---|---|---|---|
| **Individual-level** | | | | |
| **Age of the mother** | Yes | No | | |
| 15–24 years | 293 | 730 | 1.00 | 1.00 |
| 25–34 years | 398 | 1534 | 0.62 (0.51, 0.75) | **0.61(0.48, 0.79)**\*\*\* |
| ≥35 years | 182 | 711 | 0.60(0.48,0.77) | **0.66(0.47, 0.95)**\*\* |
| **Religion** | | | | |
| Orthodox | 350 | 1180 | 1.00 | 1.00 |
| Protestant | 158 | 724 | 1.19(0.91, 1.56) | 0.89(0.63, 1.27) |
| Muslim | 343 | 981 | 1.36(1.08, 1.71) | 0.83(0.58, 1.19) |
| Other | 22 | 90 | 1.04(0.58, 1.86) | 0.77(0.43, 1.39) |
| **Marital status** | | | | |
| Married or living with a partner | 845 | 2802 | 1.00 | |
| Other* | 29 | 172 | 0.85(0.58, 1.24) | |
| **Education of women** | | | | |
| No education | 560 | 1913 | 1.00 | 1.00 |
| Primary | 260 | 875 | 0.97(0.81, 1.18) | 0.98(0.78, 1.24) |
| Secondary | 34 | 134 | 1.14(0.81, 1.60) | 1.26(0.82, 1.93) |
| Higher | 21 | 51 | 0.67(0.36,1.21) | 1.37(0.68, 2.77) |
| **Wealth index** | | | | |
| Poor | 412 | 1322 | 1.00 | 1.00 |
| Medium | 202 | 705 | 0.68 (0.53,0.87) | 0.99(0.74, 1.31) |
| Rich | 260 | 947 | 0.58 (0.47,0.72) | 1.09(0.80, 1.48) |
| **Employment status** | | | | |
| Not working | 533 | 1527 | 1.00 | 1.00 |
| Working | 341 | 1447 | 0.75(0.63,0.90) | 0.87(0.72, 1.05) |
| **Husband education** | | | | |
| No education | 404 | 1330 | 1.00 | 1.00 |
| Primary | 351 | 1185 | 0.86 (0.71,1.05) | 1.01(0.82, 1.24) |
| Secondary | 55 | 208 | 1.08 (0.80,1.47) | 1.29(0.92, 1.82) |
| Higher | 35 | 79 | 0.85 (0.58,1.23) | 1.07(0.69, 1.66) |
| **Birth interval** | | | | |
| ≤23 month | 90 | 411 | 1.00 | 1.00 |
| ≥24 month | 594 | 1992 | 0.96(0.76, 1.23) | |
| **Contraceptive use** | | | | |
| No | 551 | 1772 | 1.00 | 1.00 |
| Yes | 322 | 1203 | 0.74(0.61, 0.89) | 0.88(0.71, 1.08) |
| **Parity** | | | | |
| 1–2 | 337 | 1016 | 1.00 | 1.00 |
| 3–6 | 387 | 1383 | 0.89 (0.74, 1.07) | 1.08(0.84, 1.40) |
| ≥7 | 150 | 575 | 0.76 (0.59, 0.98) | 0.87(0.60, 1.26) |
| **PNC** | | | | |
| No | 826 | 2785 | 1.00 | 1.00 |
| Yes | 47 | 190 | 0.85(0.59, 1.21) | |
| **Community conversion** | | | | |
| No | 544 | 1674 | 1.00 | 1.00 |
| Yes | 330 | 1300 | 0.88(0.74, 1.06) | 0.95(0.79, 1.15) |
| **Media exposure** | | | | |

*(Continued)*

**Table 3.** (Continued)

| Variables | Under-weight | | Crude OR (95%CI) | Adjusted O(95%CI)# |
|---|---|---|---|---|
| No | 605 | 2035 | 1.00 | 1.00 |
| Yes | 269 | 939 | 0.75(0.62, 0.91) | 0.95(0.76, 1.20) |
| **Toilet facility** | | | | |
| No | 381 | 1108 | 1.00 | 1.00 |
| Yes | 493 | 1866 | 0.54(0.45,0.65) | **0.67(0.54, 0.83)**\*\*\* |
| **Hand washing facility** | | | | |
| No | 407 | 1349 | 1.00 | 1.00 |
| Yes | 466 | 1626 | 0.81(0.68,0.96) | 0.98(0.81, 1.18) |
| **Source of drinking water** | | | | |
| Improved source of drinking water | 512 | 1705 | 0.92(0.76,1.11) | |
| Unimproved source of drinking water | 362 | 1270 | 1.00 | 1.00 |
| **Chat chewing** | | | | |
| No | 746 | 2568 | 1.00 | 1.00 |
| Yes | 128 | 406 | 0.84(0.61,1.16) | |
| **Alcohol drink** | | | | |
| No | 593 | 1910 | 1.00 | 1.00 |
| Yes | 281 | 1064 | 0.73(0.59,0.91) | 0.80(0.59, 1.10) |
| **Community-level variables** | | | | |
| **Region** | | | | |
| Agrarian | 823 | 2844 | 1.20(0.87, 1.65) | 0.94(0.65, 1.36) |
| Pastoralist | 40 | 75 | 2.35(1.62, 3.43) | 1.30(0.87, 1.92) |
| City dwellers | 11 | 55 | 1.00 | 1.00 |
| **Place of residence** | | | | |
| Urban | 51 | 298 | 1.00 | 1.00 |
| Rural | 822 | 2677 | 1.87(1.40,2.51) | **1.65(1.13, 2.41)**\*\*\* |
| **Community poverty level** | | | | |
| Low | 496 | 1848 | 1.00 | 1.00 |
| High | 378 | 1126 | 1.94(1.58, 2.38) | **1.40(1.08, 1.82)**\*\*\* |

\*\*\*p-value < 0.001

\*\*p-value< 0.05; OR: Odds Ratio

**#Model-III:** a model with both individual and community-level factors

nationally composed of nine (9) regional and two (2) city administrations. At the same time, those surveys are localized to specific districts, which might over or underestimate the prevalence.

On the other hand, this is higher than the prevalence of underweight among lactating mothers previously conducted in Ethiopia, 19.5% [34], 17.4% [15], and 17.7% [35]. This discrepancy might be due to the attention given by the Federal Ministry of Health to maternal health and nutrition and engagement of NGOs in the issues of nutrition in Moyle district. Besides, the farmers produced income-generating fruit crops (banana, mango) in Arba Minch Zuria, thus were less disadvantaged.

In the current study, it was found that the odds of underweight among lactating mothers aged between 25 and 34 years and greater than 35 years were the lower chance of being underweight by 39% and 34% compared with those lactating mothers who had 15–24 years of age, respectively. This finding is consistent with a study conducted in India [36]. In another study in Ethiopia, the highest proportion of malnourished women was observed in the youngest age group [37].

The potential reasons for this might be in adolescence, and young women's nutritional needs increase because of the rapid growth that accompanies puberty and the increased demand for iron associated with the onset of menstruation [38]. According to the WHO guidelines, adolescent mothers have greater nutritional requirements than adults; due to their physical growth [39, 40].

Lactating who have toilet facilities were 37% times less likely to be undernourished compared to their counterparts. This finding is supported by the study conducted in Arba Minch Zuria [21] and Adama district [18]. The reason might be open defecation, as a lack of toilet facilities increases the risk of diarrheal illness, which can lead to underweight. The most common source of diarrheal pathogens is human feces [41]. Moreover, a household's economic condition is a predictor of access to adequate food supplies, health services, improved water sources, and sanitation facilities, all of which are important determinants of maternal nutritional status. Besides, it was also reported that households without toilet facilities are often impoverished and live in unsanitary conditions [37].

Those lactating women living in rural areas were 1.65 times more likely to be underweight than those living in urban. This finding was comparable with other studies conducted in Shashemene Woreda [16]. This difference is because Lactating women in urban areas have nutrition knowledge, which would improve their nutritional status and provide them with nutrition information on the intergenerational consequences of nutritional deficiencies. Research indicated urban women with primary education were less likely to be affected by chronic undernutrition [1, 42].

The odds of being underweight among lactating mothers who reside in high community poverty were increased by 40% compared with those who reside in lower community poverty levels. This might be because socioeconomic hardship was related to reduced media exposure or reduced educational level achievement, which decreases the likelihood of having nutrition knowledge [43, 44]. Furthermore, more than 37 percent of the Ethiopian population belongs to the poorer and poorest wealth quintiles, indicating that a large percentage of women cannot afford diversified food groups at the market area due to low socioeconomic status [14]. Moreover, as evidence shows that in underdeveloped nations, maternal undernutrition is often linked to female illiteracy, poverty, a lack of women's empowerment, and a lack of access to health care (both antenatal and postnatal care) [45, 46].

## Strengths of the study

Firstly, the data used in this analysis was the most recent, nationally representative, and a large sample of a population-based survey covering all regions and city administrators. Secondly, the DHS surveys are similar in design, having standard variables that are comparable across settings. Hence the finding could be generalized to other developing countries. Thirdly, this study is unique because it used an advanced statistical technique called multilevel modeling analysis to identify determinants of underweight among lactating mothers. This technique considers the nested nature of the DHS data, allowing for the clustering effect of the outcome variable to be examined, which is an important phenomenon to consider. Despite these strengths, the study has few limitations mentioned below.

## Limitations of the study

The findings from this study were not without limitations and should be noted. The analyses were conducted using data collected by a cross-sectional survey, which further creates a problem of making causal inferences, and due to the secondary nature of the data, the present study

was limited by unmeasured confounders such as household food insecurity, lactating women's dietary intake, monthly household expenditure, and nutrition knowledge.

## Conclusions

Both individual-level (age and toilet facility) and community-level (residence and community poverty) factors were statistically significant predictors of underweight among lactating mothers in this study. Accordingly, being rural residents and residing in high community poverty were the factors that increase the odds of being underweight among lactating mothers. In contrast, lactating mothers aged between 25 and 34 years and greater than or equal to 35 years and those who have toilet facilities decrease the odds of underweight. The findings indicate that some of the modifiable factors should be considered in the existing interventions to improve lactating mothers' nutritional status in Ethiopia.

## Supporting information

**S1 Table. The STROBE reporting items for cross-sectional study.**
(PDF)

## Acknowledgments

We are grateful to The DHS Program for allowing us to use the EDHS data for this study.

## Author Contributions

**Conceptualization:** Zinash Teferu, Yohannes Tekalegn.

**Data curation:** Zinash Teferu.

**Formal analysis:** Zinash Teferu, Yohannes Tekalegn, Biniyam Sahiledengle, Kenbon Seyoum.

**Methodology:** Zinash Teferu, Yohannes Tekalegn, Biniyam Sahiledengle, Kenbon Seyoum.

**Software:** Zinash Teferu.

**Validation:** Zinash Teferu, Yohannes Tekalegn, Biniyam Sahiledengle, Demisu Zenbaba, Fikreab Desta, Kenbon Seyoum, Habtamu Gezahegn, Damtew Solomon Shiferaw, Ayele Mamo, Vijay Kumar Chattu.

**Visualization:** Zinash Teferu, Yohannes Tekalegn, Biniyam Sahiledengle, Demisu Zenbaba, Fikreab Desta, Kenbon Seyoum, Habtamu Gezahegn, Damtew Solomon Shiferaw, Ayele Mamo, Vijay Kumar Chattu.

**Writing – original draft:** Zinash Teferu, Yohannes Tekalegn.

**Writing – review & editing:** Zinash Teferu, Yohannes Tekalegn, Biniyam Sahiledengle, Demisu Zenbaba, Fikreab Desta, Kenbon Seyoum, Habtamu Gezahegn, Damtew Solomon Shiferaw, Ayele Mamo, Vijay Kumar Chattu.

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
