## [Decision Letter · Decision Letter 0]

14 Dec 2021

PONE-D-21-26990Individual and community-level determinants of underweight among lactating mothers in Ethiopia: a multilevel analysisPLOS ONE

Dear Dr. Engida,

Thank you for submitting your manuscript to PLOS ONE. After careful consideration, we feel that it has merit but does not fully meet PLOS ONE’s publication criteria as it currently stands. Therefore, we invite you to submit a revised version of the manuscript that addresses the points raised during the review process.

We look forward to receiving your revised manuscript.

Kind regards,

Bijaya Kumar Padhi, PhD, MPH

Academic Editor

PLOS ONE

Journal Requirements:

- https://bmjopen.bmj.com/content/10/3/e034963

The text that needs to be addressed involves parts of the Discussion and Conclusions.

In your revision ensure you cite all your sources (including your own works), and quote or rephrase any duplicated text outside the methods section. Further consideration is dependent on these concerns being addressed**.**

Additional Editor Comments (if provided):

We have received reviews from two reviewers for your manuscript “Individual and community-level determinants of underweight among lactating mothers in Ethiopia: a multilevel analysis”. This is a well-written article; however, some areas need improvements. Please include a paragraph on following two key terminologies in methods section.

1) community poverty: This needs a clear description.

2) multilevel analysis: The basis of selecting multilevel analysis. How was the model built? Is it based on a priori hypothesis?

3) Please include a STROBE check list.

Reviewers' comments:

Reviewer's Responses to Questions

**Comments to the Author**

1. Is the manuscript technically sound, and do the data support the conclusions?

Reviewer #1: Yes

Reviewer #2: Yes

2. Has the statistical analysis been performed appropriately and rigorously? 

Reviewer #1: Yes

Reviewer #2: Yes

3. Have the authors made all data underlying the findings in their manuscript fully available?

Reviewer #1: No

Reviewer #2: Yes

4. Is the manuscript presented in an intelligible fashion and written in standard English?

Reviewer #1: Yes

Reviewer #2: Yes

5. Review Comments to the Author

Reviewer #1: Good quality article using secondary data. Can be further used as hypothesis for testing by planning an analytical study to prove the causal association of the variables identified. The study needs to be uptaken for intervention and policy planning and recommendations.

Reviewer #2: paper is technically fine. analysis is fine and in line with study objectives. Study also opens the door for future research in nutrition sector in Ethiopia. However, correlation and synchronization with programmatic updates will make it more useful.

6. PLOS authors have the option to publish the peer review history of their article (what does this mean?). If published, this will include your full peer review and any attached files.

Reviewer #1: **Yes: **Tanvir Kaur Sidhu

Reviewer #2: No

---

## [Author Response · Author response to Decision Letter 0]

25 Mar 2022

Dear journal editor, The Ethiopian Demographic and Health Survey was conducted after the approval of the Ethiopian National Research Ethics Review Committee. Permission to use the 2016 EDHS database for further analysis was sought from http://www.dhsprogram.com and no ethics committee approval was necessary. The data was analyzed and reported in aggregate; household and individual identifiers were not reported in the dataset. Thanks!

---

## [Editor Report · Decision Letter 1]

18 Apr 2022

Individual and community-level determinants of underweight among lactating mothers in Ethiopia: a multilevel analysis

PONE-D-21-26990R1

Dear Dr. Engida,

We’re pleased to inform you that your manuscript has been judged scientifically suitable for publication and will be formally accepted for publication once it meets all outstanding technical requirements.

Kind regards,

Bijaya Kumar Padhi, PhD, MPH

Academic Editor

PLOS ONE

---

## [Editor Report · Acceptance letter]

26 Apr 2022

PONE-D-21-26990R1 

Individual and community-level determinants of underweight among lactating mothers in Ethiopia: a multilevel analysis 

Dear Dr. Engida:

I'm pleased to inform you that your manuscript has been deemed suitable for publication in PLOS ONE. Congratulations! Your manuscript is now with our production department. 

Kind regards, 

on behalf of

Dr. Bijaya Kumar Padhi 

Academic Editor

PLOS ONE